# Molecular Characterization of the Extracellular Domain of Human Junctional Adhesion Proteins

**DOI:** 10.3390/ijms22073482

**Published:** 2021-03-27

**Authors:** Christopher Mendoza, Sai Harsha Nagidi, Dario Mizrachi

**Affiliations:** 1Department of Physiology and Developmental Biology, College of Life Sciences, Brigham Young University, Provo, UT 84602, USA; cmendoz1@vols.utk.edu; 2Department of Molecular Microbiology, College of Life Sciences, Brigham Young University, Provo, UT 84602, USA; nagidiharsha@gmail.com

**Keywords:** junctional adhesion molecules, tight junction, adherens junction, cadherin

## Abstract

The junction adhesion molecule (JAM) family of proteins play central roles in the tight junction (TJ) structure and function. In contrast to claudins (CLDN) and occludin (OCLN), the other membrane proteins of the TJ, whose structure is that of a 4α-helix bundle, JAMs are members of the immunoglobulin superfamily. The JAM family is composed of four members: A, B, C and 4. The crystal structure of the extracellular domain of JAM-A continues to be used as a template to model the secondary and tertiary structure of the other members of the family. In this article, we have expressed the extracellular domains of JAMs fused with maltose-binding protein (MBP). This strategy enabled the work presented here, since JAM-B, JAM-C and JAM4 are more difficult targets due to their more hydrophobic nature. Our results indicate that each member of the JAM family has a unique tertiary structure in spite of having similar secondary structures. Surface plasmon resonance (SPR) revealed that heterotypic interactions among JAM family members can be greatly favored compared to homotypic interactions. We employ the well characterized epithelial cadherin (E-CAD) as a means to evaluate the adhesive properties of JAMs. We present strong evidence that suggests that homotypic or heterotypic interactions among JAMs are stronger than that of E-CADs.

## 1. Introduction

Tight junctions (TJs) are cell–cell promoting structures localized to the apical region of endothelial and epithelial cells. TJs function as barriers, controlling the paracellular space, and forming an apical/basolateral intramembrane diffusion barrier in the outer leaflet of the plasma membrane, referred to as the fence function [1,2]. Dysfunction of the TJ is relevant to edema, jaundice, diarrhea, inflammatory bowel disease, and metastasis among others conditions [3,4,5,6]. TJs are proteic structures represented by a complex mixture of three membrane proteins: claudins (CLDNs), occludin (OCLN), and junctional adhesion molecules (JAMs). Additionally, adapter and effector proteins anchor the TJ to the cytoskeleton, indicating its relevance in mechanotransduction [7,8]. The role of CLDN, compared to that of OCLN, appears to be crucial for the barrier function of TJs [7,8,9,10]. The role of JAMs in controlling permeability has been discussed in terms of its function as a gatekeeper to prevent ions or molecules such as water from crossing through the paracellular space [11]. JAMs are members of the immunoglobulin super family. Four members of the JAM family have been identified as members of TJs: JAM-A, JAM-B, JAM-C and JAM 4 [12,13,14,15].

JAMs are important in the control of vascular permeability and immune cells transmigration across endothelial–cell barriers by engaging in homotypic, heterotypic interactions [16]. JAM-A interactions assure strong cell–cell adhesion, playing important roles in proliferation and epithelial cell barrier functions [17]. Alterations to the barrier integrity caused by the disruption of JAM-A can indirectly modulate immune responses [18]. JAMs are expressed by a large variety of cell types and tissues, including epithelial and endothelial barriers, cells of the male reproductive system, and cells of the central and peripheral nervous systems [19]. Recent studies of JAM-A and JAM-C have expanded their roles to include tumorigenic functions, the inhibition of apoptosis and promoting proliferation, and epithelial-to-mesenchymal transition [20,21]. JAM proteins’ relevance in cell and tissue physiology, and pathophysiology, may be obscured by the lack of characterization of their interactions.

The purification of JAM proteins has been a bottleneck in their structural characterization due in part to their transmembrane domain. However, employing the extracellular domain of JAM-A, Prota et al., 2013 [22] successfully obtained a crystal structure. This study provided information of the quaternary structure of JAM-A, a homodimer, linked through its first extracellular immunoglobulin domain. However, there is a gap of understanding in the specific case of JAM-B, -C or 4. Additionally, heterotypic interactions of JAMs may play a relevant role in physiological events [19]. Nevertheless, there are no literature reports regarding these heterotypic interactions, remaining unclear if they are energetically favored in nature, and their possible role for cellular and tissue physiology.

To address this gap in knowledge requires that we purify all JAM proteins while maintaining their native adhesive properties, in order to determine, through structural studies, their oligomerization state, as well as homotypic and heterotypic interactions. Here, we present such a study, where through maltose-binding protein (MBP) fusion strategies we have purified all extracellular domains of the TJ’s JAM proteins. This strategy enabled yields of sufficient quantities of material for protein–protein interactions studies, circular dichroism and surface plasmon resonance.

## 2. Results and Discussion

In the classical sense, cell adhesion is classified under several subcategories. Thus, cell–cell interactions and cell–basal membrane anchorage are examples of well recognized fields of study. Proteins responsible for cell–cell adhesion are membrane proteins. Most membrane proteins are naturally of low abundance and require a unique platform for expression and purification [23]. However, yields of proteins with a native-like structure and function following overexpression in bacteria, yeast, insect cells or cell-free systems are often still inadequate. Protein engineering techniques, for example those employing fusion partners, are used to increase expression and stability [24]. In the case of adhesion molecules or other membrane proteins that have a single transmembrane helix, a reliable strategy is to study the intracellular or extracellular domains separately [25]. A classic example is that of the cadherin family of cell adhesion molecules [26]. Epithelial cadherin (E-CAD) has been studied by multiple biophysical methods [27]. E-CAD serves as an excellent standard for cell adhesion since its constant of affinity (K_D_) has been well established [28]. JAM proteins are also cell adhesion proteins, a subfamily of the immunoglobulin superfamily. JAM4 has only been reported recently [15]. JAMs are integral part of TJs. Their structural difference, when compared to CLDN or OCLN, both 4-α helix membrane proteins, is striking. How do these structurally different proteins interact with each other? How do they support the structure and function of the TJ? Our research strategy aims to elucidate the structural properties of JAMs that may qualify these proteins to be part of the only protein structure that controls the paracellular space [29].

### 2.1. JAM Expression and Purification in Esherichia coli

Web-based amino acid sequence analysis (see Section 3) was key in the design of a single expression strategy for all JAM proteins, later extended to E-CAD, for consistency. We confirmed that all the members of the JAM family in humans have two conserved immunoglobulin folds in the extracellular domain, in spite of the low amino acid sequence homology (less than 15%) (Appendix A). In the JAM family, JAM-A was the most hydrophilic molecule while JAM4 was the most hydrophobic. JAM4 is also the protein that presents the most disordered regions, especially in the cytosolic region (Appendix A). Protein modeling also revealed similarities in the extracellular domain of these proteins (Appendix A).

Following the in silico evaluation of the extracellular domain of human JAMs, we examined the direct expression of all the targets. JAM-B and JAM-C were successfully expressed with low yields compared to the other proteins in this study (Appendix A). Other designs contained a TEV protease cleavage site between MBP and the JAM, but unfortunately resulted in mixed species and very low yields (data not shown). Our final strategy was the use of plasmid pET28-MBP, with MBP N-terminal to the gene of interest (Appendix A). This was based on the merits described in the literature to drive high protein expression and stability of the fused target, enabling the targets to retain their individual structure and function [30,31]. The pET28-MBP was subcloned to contain the extracellular domains of JAM-A, JAM-B, JAM-C, JAM 4 and E-CAD (see Section 3 and Appendix A). To produce high yields of proteins and allow proper disulfide bond formation and cytosolic expression, we used the SHuffle T7 bacterial strain [32].

Cell growth was monitored for a 24-h period (Appendix A). Plasmids hosting JAM targets hampered the growth of the cells at 37 °C in LB containing both kanamycin (required by pET28-MBP) and spectinomycin (required by SHuffle cells) [32]. Bacterial growth was determined to have to reach an OD_600_ of 0.8–1 the before addition of 0.1 M of IPTG. Culture continued at 16 °C overnight. Protein purification with amylose resin, followed by size exclusion, produced sufficient yields of >95% pure protein for structural studies (Figure 1B).

### 2.2. Size Exclusion Chromatography

Size exclusion chromatography identified a unique feature among JAMs, their quaternary structure. E-CAD formed dimers as described in the literature [33], similar to the published oligomeric state of JAM-A [22]. This led to the determination of whether the other JAMs (-B, -C and 4) had a similar oligomeric state. Here, we report for the first time that this is not the case. JAM-C, JAM-B and JAM 4 form higher orders of organization. Through size exclusion, we determined that JAM-C forms tetramers, JAM-B forms octamers, and JAM4 forms decamers (Figure 1D). Our research strategy did not identify the exact organization of these oligomers. Nevertheless, considering that the basic organization for adhesion molecules is *trans* interactions, we suggest that JAMs may form a higher order oligomerization on *trans* dimers [22,33].

### 2.3. Determination of Conserved Secondary Structures by Circular Dichroism (CD)

Based on the dimerization results obtained in size exclusion chromatography, we decided to determine whether the JAM proteins shared a conserved secondary structure. In Figure 2, we present two pieces of evidence that seem to indicate that beyond the in silico analysis and protein modeling, the extracellular domain of JAMs is composed of high β-sheet structures. The crystal structure of MBP [34], as well as CD [35] data, found in the literature, and performed at 21 °C has been reported to have an α-helix content of 36%, and a β-sheet content of 17%. We used these values to compare our results. Our CD data indicates that under the conditions of our experiment (22 °C), MBP has 38.1% of α-helix content and 19.1% of β-sheet content, consistent with the literature. In Figure 2A, we plotted the CD data for all MBP-fused extracellular domains. This graph indicates that these proteins, with their immunoglobulin domain render similarities in the final fusion protein composed of MBP and JAMs. The CD values were compared according to the percentage of α-helix, and β-sheet. Figure 2B shows that these fusion proteins had a higher β-sheet content compared to the unfused MBP at only 19.1%. MBP-fused JAMs were examined and compared to MBP alone. JAM-A increased the β-sheet content of the fusion to over 50%, which could be the result of the increased thermal stability of JAM-A (Appendix A). The least thermal stable JAM-C (Appendix A) influenced its fusion with MBP to adopt to a greater coil–coil structure (Figure 2B). On the other hand, the extracellular domain of JAM 4 greatly increased both the α-helix and β-sheet content of the fusion. This could be due to the hydrophobicity of JAM 4 (Appendix A) or its quaternary organization (Figure 1D) or a combination of both.

Based on both the size exclusion chromatography and CD, we asked whether based on the difference in aggregation and of secondary structures these proteins produced tighter binding in homotypic or heterotypic interactions. To address this question, we employed surface plasmon resonance (SPR). This technique is used to measure the binding of molecules in real-time without the need of labels [37,38]. Using this technique, we determined both the homotypic and heterotypic interactions of JAM-A, JAM-B and JAM-C that have been reported [19,38,39,40] and compared it to JAM4, which remains understudied.

### 2.4. Homotypic Interactions of JAMs

Vendome and colleagues [27], discussed the formation of homotypic interactions in the case of E-CAD. While exploring affinity constants obtained by multiple methods (SPR and analytical ultracentrifugation) the authors conclude that rather than obtaining absolute values, they observed that the data are in agreement of the behavior of E-CAD. Vendome and colleagues explain that each technique offers enough differences to produce unequal values even when measuring the same properties of the same protein [27]. Our research strategy circumvented this paradigm by measuring both E-CAD and JAMs’ protein–protein interactions using SPR, with E-CAD values serving as a standard parameter. Finally, rather than interpret the absolute value determined by SPR, we normalized the values presented (affinity constant, K_D_) to that of the better studied E-CAD. In addition to the comparison of protein–protein interactions, normalizing K_D_ to that of E-CAD enables a simple estimation of the adhesion contributions of the AJ and the TJ to the intercellular interactions.

Figure 3 offers the normalized affinity values (K_D_) for the members of the JAM family (see also Table 1). Compared to E-CAD vs. E-CAD (in the presence of Ca^+2^) all JAM proteins displayed a higher affinity (K_D_) for the homotypic interactions. JAM4 demonstrated over 1000-fold higher affinity than E-CAD. JAM-A, -B, and -C presented 5-, 25-, and 8-fold stronger affinity than E-CAD.

### 2.5. Heterotypic Interactions of JAMs

When cells expressing more than one JAM protein in the TJ establishes contact with a neighboring cell, a variety of heterotypic interactions between JAMs may occur [19]. If JAMs interact in a *cis* fashion, we would not see an interaction with SPR. However, if JAMs interact in a *trans* fashion, the proteins involved would interact with SPR. Based on our SPR data, we were able to determine that JAMs interact in a *trans* fashion. Sodium caprate is a detergent known for disrupting the TJ and increasing the paracellular permeability [41]. Our SPR experimental design included the use of caprate to eliminate the *trans* interactions (mimicking the effects of caprate in vitro and in vivo) [41]. Figure 4 offers the normalized affinity values (K_D_) for the members of the JAM family (see also Table 1). Each graph is normalized according to the homotypic interactions of the corresponding JAM.

Figure 4A suggests that heterotypic interactions between JAM-A and JAM-B or JAM4 may be favored over JAM-A homotypic interactions. This corresponds with reports where JAM-A and JAM-B interact during embryonic development [13]. In the case of JAM-B, its preferred heterotypic interaction should be with JAM-C (Figure 4B). JAM-B and JAM-C regulate the migration of cerebellar granule neurons during development of the cerebellum [42]. Even though they have not been reported to form heterotypic interactions in vivo or in vitro, our data suggest that these heterotypic species may play a key role in the brain. Both JAM-C and JAM4 experiments (Figure 4C,D) show that the recorded data suggest these interactions are highly favored. Unfortunately, the lack of research regarding JAM4 makes it difficult to further evaluate the observed results. Finally, JAM-A and JAM-C are expressed on the surface of platelets [43]. Their role in the coagulation cascade is unclear. Considering that platelets under homeostatic conditions do not aggregate, data collected here seem to be in agreement that JAM-A and JAM-C may not interact with each other if platelet aggregation was triggered. Their participation in the coagulation cascade might yet remain related to their adhesive properties but involving other proteins, for example, integrins [12].

Our study provides insight into the vast interactions of JAM proteins. It is not surprising that the JAM protein family performs homotypic and heterotypic interactions. The dimerization of JAM-A was validated in the crystal structure obtained by Prota [22]. Here, we determined that there are different oligomeric states formed by JAM-B, JAM-C and JAM 4. Specifically, we found that JAM 4, the most hydrophobic member of the family, forms a higher quaternary order, compared to the least hydrophobic JAM-A, that forms a dimer. These results suggest that there is a difference in binding between these proteins, and that promiscuous interactions among other members of the family may be equally relevant.

We present evidence that the JAM proteins form homotypic and heterotypic interactions with members of the JAM family. Unlike previously published work, we have seen that the heterotypic interactions tend to have a lower K_D_ value, suggesting that there is a tighter binding. Our data indicate that the formation of heterotypic interactions may be more favored when compared to homotypic interactions. The relevance of our findings could indicate that JAMs play a major role in controlling the paracellular space, and thus tissue barriers. JAMs may also play a key role in hemostasis.

Interestingly, we confirmed through circular dichroism that these proteins share a similar secondary structure. Furthermore, this could be crucial for the function of these proteins in the tight junctions. The hydrophobic profiles of JAMs demonstrated a striking difference among members; these can be a major difference in oligomeric assembly in both homotypic and heterotypic interactions. Circular dichroism revealed that all of these proteins lost their ability to retain secondary structure, and did not fold at 50 or 75 °C. This is due to the breakdown of hydrophobic and Van der Waals interactions. This result indicates that these proteins would fold correctly at physiological temperature, while at higher temperatures they would become dysfunctional.

Finally, our work might shed light on the fundamental question of why there are four different JAM proteins and what their specific roles are in the formation of tight junctions in specific tissues. The tissue-specific expression of JAMs is only partially known, but well established in other tissues. Our evidence suggests a greater role of JAMs in permeability than previously suggested [40,44]. Based on our findings, heterotypic interactions could result in stronger intercellular interactions, leading to further control of the TJ permeability, thus conforming to the tissue homeostatic needs. We might imagine a scenario in which there needs to be an interaction of the most hydrophobic JAMs, either homo- or heterotypic, to regulate permeability in tissues such as the blood–brain barrier. Such a scenario can be responsible for the trafficking of ions, water and other hydrophilic molecules through the tight junction barrier. Opposite to the blood–brain barrier is the case of the kidney, where certain regions support the reabsorption of ions and water. In this case, control of the tight junction’s permeability may rely on JAM expression of the less hydrophobic JAMs, such as JAM-A. Future work would investigate these ideas, and what is clear from our research is that there is a difference in how these JAMs oligomerize, and that they form homotypic and heterotypic interactions. Understanding these differences may result in unveiling the specific inner workings in tight junctions and its control of the paracellular permeability. In Figure 5, we summarize the homotypic and heterotypic interactions of JAMs, and rank them according to their strength, or in other words, the strong adhesive contributions to the tight junction.

## 3. Materials and Methods

### 3.1. Materials

All cloning and PCR reagents were obtained from New England Biolabs (Ipswich, MA, USA. https://www.neb.com/, accessed on 10 March 2021). Amylose resin was purchased from NEB and used according to manufacturer’s protocol. All chemicals were purchased from Sigma–Aldrich (St. Lois, MO, USA. https://www.sigmaaldrich.com/united-states.html, accessed on 10 March 2021). pET28a empty vector was obtained from Sigma–Aldrich, catalog number 69864. pMAL c2x plasmid (discontinued from New England Biolabs) was used to generate maltose binding protein (MBP) as a gene of interest to clone into pET28a between restriction sites NcoI and NdeI (Appendix A).

### 3.2. Web-Based Analysis Tools

Amino acid sequence alignment was performed using phylogeny, https://www.phylogeny.fr/, accessed on 10 March 2021, which uses a MUSCLE amino acid sequence alignment. The generation of hydropathy plots was carried out using Expasy ProtScale: https://web.expasy.org/protscale/, accessed on 10 March 2021. The order, and disorder plots were obtained using: http://www.pondr.com/, accessed on 10 March 2021. Bestsel circular dichroism analysis was carried out using http://bestsel.elte.hu/index.php, accessed on 10 March 2021.

### 3.3. Protein Models

Models and molecular graphics images were produced using the UCSF Chimera v. 1.15 package from the Resource for Biocomputing, Visualization, and Informatics at the University of California, San Francisco (supported by NIH P41 RR-01081) [45].

### 3.4. Cloning, Expression and Purification of Proteins

gBlocks for the extracellular domains of human proteins JAM-A (Gly27-Arg228, accession number Q9Y624), JAM-B (Gly28-Ser238, accession number P57087), JAM-C (Gly30-Glu236, accession number Q9BX67), JAM4 (Gly34-Arg286, accession number Q9NSI5), and E-CAD (Val102-Asp312, accession number P12830) were obtained from IDT DNA Technologies (Iowa City, IA, USA. https://www.idtdna.com/pages, accessed on 10 March 2021) (Appendix A), codon optimized for *E. coli K-12* (IDT DNA Technologies Codon Optimization Tool). The gBlocks were amplified with forward and reverse primers (Appendix A), followed by restriction enzyme digestion (XhoI and NdeI, New England Biolabs). Fragments were subcloned in pET28a-MBP plasmid, kanamycin resistant (Appendix A). The final product produces an N-terminal MBP-fusion protein of the target with a C-terminal 6xHis tag. Cloning and subcloning transformations were performed in NEB 5-alpha (New England Biolabs). Plasmids for protein expression were transformed into SHuffle T7 Express (New England Biolabs), spectinomycin resistant. Protein expression and purification (amylose resin) were performed following manufacturer’s instructions. Eluate was concentrated by using Microsep Advance with 10k Omega centrifugal devices from Pall Corporation (Port Washington, NY, USA. https://www.pall.com/, accessed on 10 March 2021).

### 3.5. Size Exclusion Chromatography (SEC)

All size exclusion chromatography was performed using the NGC Chromatography System and its accompanying software (Hercules, CA, USA. https://www.bio-rad.com/, accessed on 10 March 2021). The SEC column used to purify proteins of interest was ENrich™ SEC 650 10 × 300 (BioRad, Hercules, CA, USA). Protein concentration was determined by using the Nanodrop One^c^ from Thermo Scientific. PBS was employed as a running buffer for SEC. Fractions were pooled and concentrated as mentioned above. Product peaks were compared for the position to the size exclusion standards from BioRad Catalog Number 151-1901.

### 3.6. Circular Dichroism (CD) Spectrometry

CD measurements were performed on the Spectrophotometer Model 420 AVIV (Biomedical Inc. Lakewood, NJ USA), calibrated with PBS. The far UV–CD spectra of 0.100 µM target protein was equilibrated with PBS (pH 7.4–8.0) and recorded in 100 mm QS glass cuvette cell. For the analysis of thermal stability and changes in the secondary structure 0.100 µM of protein sample was incubated at 4 °C, 22 °C, 37 °C, 50 °C, and 75 °C, and monitored by measuring changes in ellipticity at 260 nm to 195 nm using 20 s scans. A secondary structure consisting of alpha helix or beta sheet percentages was performed by the usage of Bestsel circular dichroism analysis, http://bestsel.elte.hu/index.php, accessed on 10 March 2021. The experimental design for CD was performed with the MBP fusion protein since there was a low yield of protein if there was a TEV cleavage site or with other constructs.

### 3.7. SDS-PAGE Assay

Two µg of either boiled or not boiled MBP, E-CAD, JAM-A, JAM-B, JAM-C and JAM4 were electrophoresed on 8% SDS-PAGE gel (BioRad). Gel staining was performed using standard protocols [46].

### 3.8. Surface Plasmon Resonance (SPR)

SPR was performed using Open SPR by Nicoya Lifesciences (Kitchener, ON, Canada. https://nicoyalife.com/, accessed on 10 March 2021). We assayed protein–protein interactions by loading 0.100 mg of each protein as ligand into the Carboxy sensor chip (Nicoya Lifesciences). The proteins are immobilized in the Carboxy sensor chip through the exposed primary amine groups that are both found on the lysine residues and the N-terminus of the amino acid residues. As a result, the amines can form a covalent bond with the carboxyl surface after it is activated by EDC/NHS [47]. Following the blocking step (manufacturer’s buffer) 200 µL of 1 M sodium caprate was administered to disrupt the preformed protein–protein interactions. All proteins analyzed formed at least dimers; these species needed to be disrupted in order to determine new protein–protein interactions kinetics. Triplicate injections of the analyte protein were performed in concentrations of 12.5 µg, 25 µg, 50 µg and 100 µg per 200 µL injections. Caprate injections were performed after each analyte interaction was concluded. The close curve fitting to the sensograms were calculated by global fitting curves (1:1 Langmuir binding model). The data were retrieved and analyzed with TraceDraw software (Kitchener, ON, Canada).

### 3.9. Surface Plasmon Resonance Statistics

SPR for each run was performed 3 times per sample, and analyzed by using the TraceDraw software (Kitcherner, ON, Canada) according to the suggestions by Nicoya (Kitchener, ON, Canada). The data in Figure 3 were normalized by using the K_D_ values of each combination run (samples) by dividing each by the E-CAD vs. E-CAD K_D_ value. For Figure 4, we normalized the heterotypic K_D_ values with that of the homotypic interaction for each JAM protein. Thus, all heterotypic interactions of JAM-A were normalized to the K_D_ value of the homotypic JAM-A interaction. We performed a similar analysis with JAM-B, JAM-C and JAM4.

## 4. Conclusions

JAMs are an integral component in the formation of TJs, but very little is known about their specific role in those TJs. In this study, we determined that it is possible for these JAM proteins to have homotypic and heterotypic interactions. Our contributions expand the understanding that each member of the JAM family may have a different quaternary organization, beyond what was previously reported for JAM-A. Here, JAM-B, -C, and 4, form tetramers and multimers. Additionally, these JAM proteins have similar secondary structures that could represent the basis of similar function. Based on these results, we propose a model where it could be possible for these proteins to interact in combinations of JAMs based on the tissue specificity and tissue environment. All authors have read and agreed to the published version of the manuscript.

## Figures and Tables

**Figure 1 ijms-22-03482-f001:**
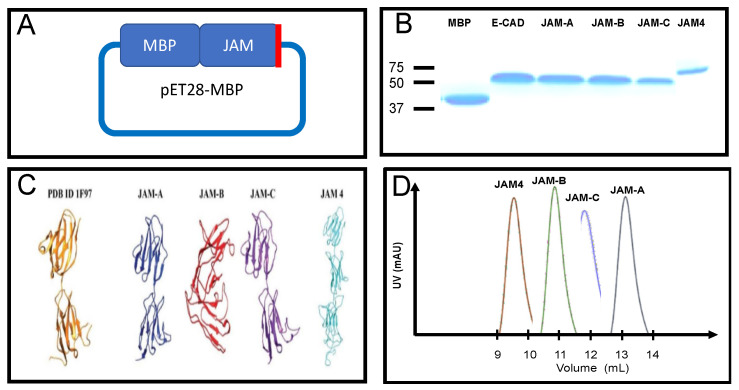
Characterization of junction adhesion molecule (JAM) family of proteins. (**A**) Extracellular domains of human epithelial cadherin (E-CAD) or JAM proteins (Appendix A) were cloned C-terminal to maltose binding protein (MBP) in pET28a backbone plasmid, the red line indicates a 6xHIS tag C-terminal to the target protein. (**B**) Proteins are purified with amylose resin and size exclusion chromatography. Proteins are purified to >95% purity, Coomassie blue stain gel. (**C**) Additional characterization was performed through in silico protein models (see Materials and Methods) of JAMs, the crystal structure of JAM-A (PDB ID: 1F97) is next to the models for comparison. (**D**) Size exclusion profiles are overlapped to show how each JAM protein forms unique quaternary structures.

**Figure 2 ijms-22-03482-f002:**
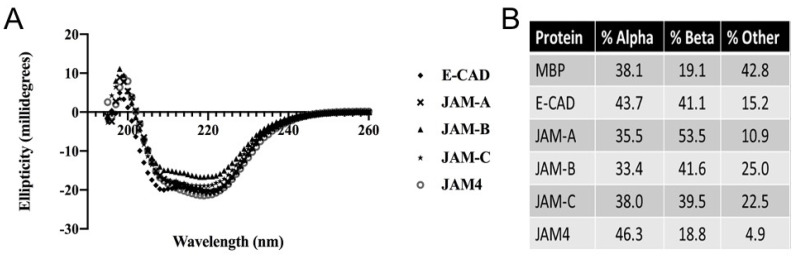
Circular dichroism analysis of JAM proteins and E-CAD. (**A**) Circular dichroism analysis, comparison of all MBP fusion, extracellular domain of E-CAD or JAM proteins. Folding similarities are observed for the behavior of the fusion proteins. (**B**) The analysis of each curve (Materials and Methods) for the target proteins, including non-fused MBP, is presented in a table that describes the distribution of secondary structure. The content of alpha helix, beta sheet or other (coiled) is presented. Our non-fused MBP protein displayed a similar distribution of the secondary structure as previously reported in the literature [35,36].

**Figure 3 ijms-22-03482-f003:**
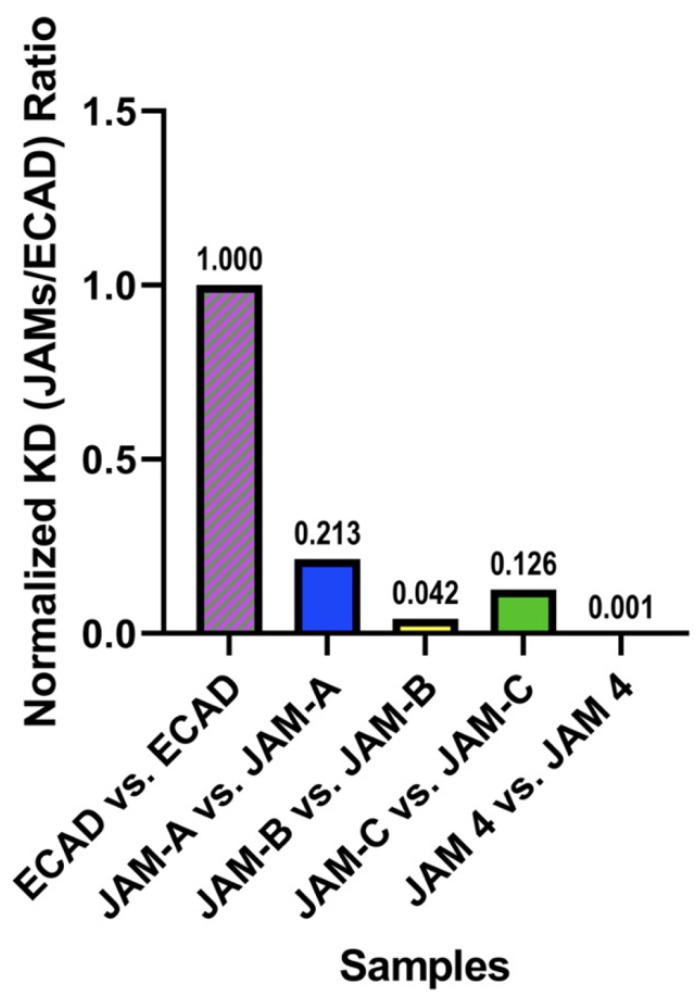
Surface plasmon resonance characterization of homotypic interactions of JAMs. Homotypic interactions of JAMs were determined by surface plasmon resonance (SPR) (see Materials and Methods). The homotypic interaction of E-CAD was also determined. Considering the large amount of evidence for E-CAD [27] we normalized the affinity (K_D_) by that of E-CAD. Thus, the Y-axis represents the normalized affinity, JAM/E-CAD as a ration. The X-axis describes the homotypic interactions tested. These values are based on taking the K_D_ values from each sample and dividing it by the K_D_ value of E-CAD vs. E-CAD shown on Table 1.

**Figure 4 ijms-22-03482-f004:**
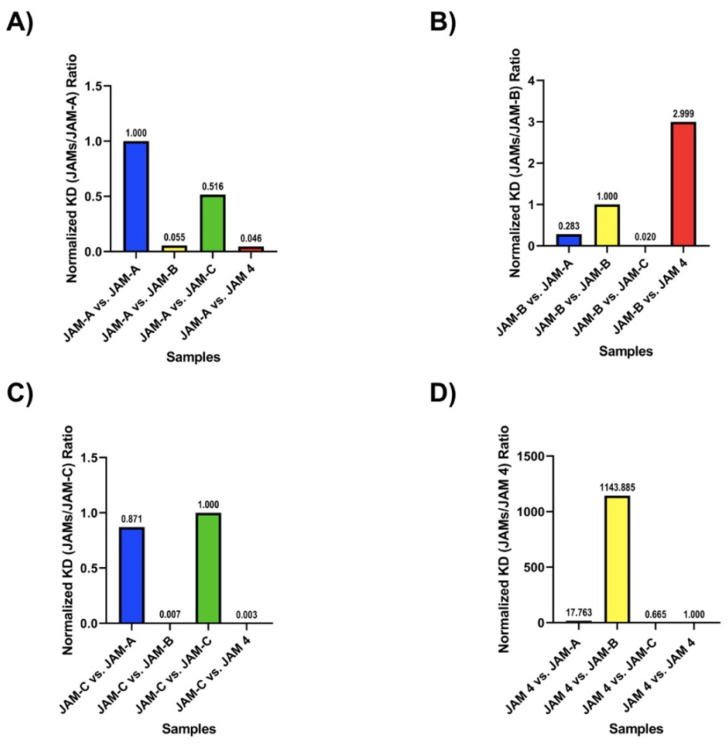
Surface plasmon resonance characterization of heterotypic interactions of JAMs. Heterotypic interactions of JAMs were determined by SPR (see Section 3 and Table 1). We studied the affinity of each JAM protein for all other members of the family. For each JAM analyzed, we normalized affinity (K_D_) by that of the homotypic interaction of said JAM. The Y-axis represents the normalized affinity, heterotypic JAM/homotypic JAM, as a ration. The X-axis describes the homotypic interactions tested. (**A**) Heterotypic interactions of JAM-A, normalized to K_D_ of JAM-A vs. JAM-A. (**B**) Heterotypic interactions of JAM-B, normalized to K_D_ of JAM-B vs. JAM-B. (**C**) Heterotypic interactions of JAM-C, normalized to K_D_ of JAM-C vs. JAM-C. (**D**) Heterotypic interactions of JAM4, normalized to K_D_ of JAM4 vs. JAM4.

**Figure 5 ijms-22-03482-f005:**
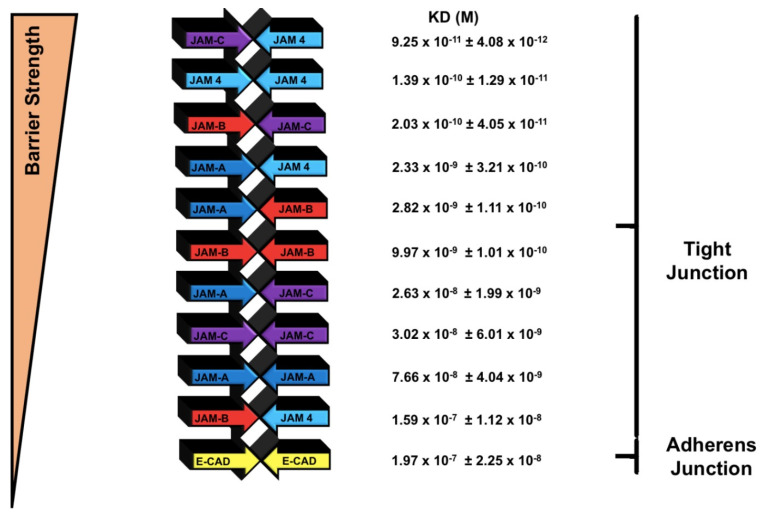
Ranked strength of homotypic and heterotypic interactions. In this figure, we summarize the major findings of our research. Graphically, we display the homo- and heterotypic interactions of JAMs, and the homotypic interactions of E-CAD. The ranking of these interactions is in order of strength.

**Table 1 ijms-22-03482-t001:** Surface Plasmon Resonance analysis. Protein–protein interactions were analyzed using SPR (see Materials and Methods). The data were analyzed with TraceDraw software. Here, we present values for constant of association, Ka (1/(M*s)); constant of dissociation, Kd (1/s); and constant of affinity, KD (M). Standard deviations are reported. * E-CAD experiments conducted in the presence of 3 mM CaCl2.

PPI Evaluated	K_a_ (1/(M*s))	K_d_ (1/s)	K_D_ (M)
E-CAD vs. E-CAD *	4.65 × 10^3^ ± 1.61 × 10^2^	6.96 × 10^−4^ ± 7.87 × 10^−5^	1.97 × 10^−7^ ± 2.25 × 10^−8^
JAM-A vs. JAM-A	1.31 × 10^4^ ± 5.30 × 10^3^	8.38 × 10^−4^ ± 2.18 × 10^−5^	7.66 × 10^−8^ ± 4.04 × 10^−9^
JAM-B vs. JAM-B	4.80 × 10^3^ ± 8.84 × 10^2^	5.36 × 10^−5^ ± 5.55 × 10^−6^	9.97 × 10^−9^ ± 1.01 × 10^−10^
JAM-C vs. JAM-C	7.82 × 10^2^ ± 1.89 × 10^1^	2.26 × 10^−5^ ± 2.99 × 10^−6^	3.02 × 10^−8^ ± 6.01 × 10^−9^
JAM4 vs. JAM4	1.67 × 10^3^ ± 4.20 × 10^2^	2.04 × 10^−7^ ± 1.64 × 10^−8^	1.39 × 10^−10^ ± 1.29 × 10^−11^
JAM-A vs. JAM-B	1.65 × 10^4^ ± 1.01 × 10^3^	5.09 × 10^−5^ ± 2.10 × 10^−6^	2.82 × 10^−9^ ± 1.11 × 10^−10^
JAM-A vs. JAM-C	2.93 × 10^4^ ± 3.31 × 10^3^	3.85 × 10^−5^ ± 6.01 × 10^−6^	2.63 × 10^−8^ ± 1.99 × 10^−9^
JAM-A vs. JAM4	1.60 × 10^3^ ± 1.00 × 10^2^	4.09 × 10^−6^ ± 1.17 × 10^−7^	2.33 × 10^−9^ ± 3.21 × 10^−10^
JAM-B vs. JAM-C	8.54 × 10^4^ ± 2.09 × 10^3^	1.76 × 10^−5^ ± 1.00 × 10^−6^	2.03 × 10^−10^ ± 4.05 × 10^−11^
JAM-B vs. JAM4	1.21 × 10^3^ ± 0.99 × 10^2^	1.28 × 10^−4^ ± 8.79 × 10^−6^	1.59 × 10^−7^ ± 1.12 × 10^−8^
JAM-C vs. JAM4	2.03 × 10^3^ ± 9.18 × 10^2^	1.96 × 10^−7^ ± 1.25 × 10^−8^	9.25 × 10^−11^ ± 4.08 × 10^−12^
JAM-A vs. E-CAD *	2.34 × 10^3^ ± 8.11 × 10^2^	7.08 × 10^−4^ ± 3.33 × 10^−5^	3.08 × 10^−7^ ± 2.02 × 10^−8^

## Data Availability

Not applicable.

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
