# Peer review of "Molecular Characterization of the Extracellular Domain of Human Junctional Adhesion Proteins"

_ijms, 2021, doi:10.3390/ijms22073482_

Round 1

Reviewer 1 Report

Sir, 

I have reviewed the manuscript "Molecular characterization of the extracellular domain of human junctional adhesion proteins" submitted to IJMS by Christopher Mendoza and co-workers. 

I find the topic of human junctional adhesion proteins highly relevant for epithelial biology. This JAM group is known for a long time, however, it was always overshadowed by more prominent molecules like claudins and occludins. 

I believe that the introduction is very/too brief. It would be worthy to extend it with perspective to clinical research application. The statement in line 28 ("edema, jaundice, diarrhea, inflammatory bowel disease, and metastasis") seems to be not very explanatory and does not ignite any interest of novice reader. I believe that this must be/ could be improved easily. 

The results are interesting and I believe that the data on tighter binding in homotypic or heterotypic interactions are a valuable contribution to further resarch. Also, the authors confirmed through Circular Dichroism that these proteins share a similar secondary structure. This is another notable achievement. 

The quality of the graphics is odd. Figure 1, panel A overlaps with line 100. It is just a matter o formatting. Figure 2, panel A id blurry and difficult to read. Higher DPI is probably needed. Also in Figure 4, the lettering is difficult to read. 

I believe that modifications are easy to do and this manuscript possesses a good potential for final approval soon. 

Author Response

Dear Reviewer,

it was an absolute delight to receive your comments. We have done much to provide the research field with new tools to examine the role of JAMs. 

As you requested we have improved our Introduction to include more information regarding the physiology and pathophysiology associated with this family of proteins. We made a few minor changes to provide the proper flow of the content.

We have additionally followed your indications regarding the figures. We have made the corresponding changes to the resolution and formatting.

We appreciate your help in making this first publication from our group about JAMs a success.

Sincerely

Dario Mizrachi

Reviewer 2 Report

General comment:

This manuscript, entitled “Molecular characterization of the extracellular domain of human junctional adhesion proteins,” authored by Mendoza et al., reports the role of the junction adhesion molecule (JAM) family of proteins in connection with their biophysical and biomolecular properties. The homotypic and heterotopic interaction plays an essential role in these proteins' class to decide the permeability in a specific tissue. This kind of approach is suitable for connecting biochemical and biophysical events to structure and conformation and is nice to answer fundamental biological questions in protein science. In my opinion, this is a valuable work and is suitable for publication in Int. J. Mol. Sci. after the authors have addressed the following comments and questions:

Specific comments:

  • Fig 1B – Whether JAM protein was expressed with MBP fusion? I yes whether it cleaved before running SDS PAGE? The minor suggestion here, please run the actual molecular weight marker in the same gel.
  • Fig 1B – The gel's molecular weight matched with the calculated weight (as in supplementary for JAM-A). What is the molecular weight for JAM4 as a band is showing here higher compare to other JAMs?
  • For size exclusion chromatography – did the author ran as molecular weight standard to calculate molecular weight? It is crucial to explain the exact stoichiometry of the complex.
  • How do you measure the alpha-helix or beta-sheet percentage – is it software-based – like CD tool? Please mention that. Visually I can see E-CAD has more alpha-helical characteristics spectra than JAM-4 – Fig 2. Explain it.
  • Please correct the values of KD (Y-axis) in figure 3 is calculated from table1. Although it has a similar pattern, values are not correct for KDJAMs/ECAD.
  • Why were experiments like CD performed with fusion proteins? MBP fusion will interfere with the actual mobility in case of size exclusion and affect secondary structure analysis. I guess the same is valid for Surface plasmon resonance.
  • In the SPR experiment, how you immobilized the JAMs proteins on carboxy sensor chips? Elaborate the method section. Which buffer and additives were used like NHS-EDC? How you confident there is no nonspecific interaction even after using blocking agents and sodium caprate? Please show the chromatogram events. If the main text has a figure limit, please add it in supplementary.

Author Response

Dear Reviewer,

it was an absolute delight to receive your comments. We have done much to provide the research field with new tools to examine the role of JAMs. 

We improved our Introduction to include more information regarding the physiology and pathophysiology associated with this family of proteins. We made a few minor changes to provide the proper flow of the content.

Directly to your comments. We have performed every experiment with the fusion of MBP. We even extended the same to E-CAD. The reason MBP was included is that the cleavable linker was not efficient and will produce mixtures. JAMs and E-CAD can form dimers and thus some of them will be MBP-JAM with JAM alone. In other cases, separating JAMs from the MBP fusion will result in a decreased product. JAM4 and JAM-C are among the most difficult targets because of their hydrophobicity. By maintaining the MBP and using it consistently in all our samples we were able to derive results that can be applied to the entire family. By comparing the results with the data from E-CAD we were able to extrapolate the strength of the binding of JAMs. It was an indirect method, because of MBP, but by normalizing the data with MBP-E-CAD we were able to validate the results. The SPR data (KD) of MBP-E-CAD was within 2-3 fold of the calculated KD in the literature.

  • Size exclusion chromatography – we included more details regarding the molecular weight standard from BioRad used to make the calculations of molecular weight.
  • The alpha-helix or beta-sheet percentage it is derived from an online source that analyzed the CD spectra. We have included the details of the online source. We have explained better the results of this analysis.
  • We have corrected the deficits of the KD in the table and Figures as you pointed out.
  • We included a citation and further explanation regarding the immobilization of the proteins for SPR.

We appreciate your help in making this first publication from our group about JAMs a success.

Sincerely

Dario Mizrachi

Reviewer 3 Report

An interesting original paper on junctional adhesions molecules molecular characterization.

only minor queries:In the materials and methods section, a subsection about statistical analysis should be added, explaining what statistical test you did perform during the experiments.

Conclusions should be expanded highlighting the possible developments subsequents to this study.

Author Response

Dear Reviewer,

we were so excited to receive your prompt response. We appreciate your diligence and support to our first contribution to the research field of the JAMs family and to the Cell-Adhesion research field. Our data indeed is relevant and enables the comparison of JAMs and also indirectly to their connections with the Adherens Junction by using E-CAD as a control. As an example, our measurements of affinity via SPR resulted in a KD within 2 to 3 fold of that calculated for E-CAD by multiple means.

We have expanded the Introduction and conclusions to reflect on the physiology and pathophysiology of these proteins and our findings. We have also included a description of the statistical analysis prior to the Normalization of the data as presented in the SPR figures. We have made clear the source of the analysis of the Circular Dichroism.

Once again, we thank your efforts and appreciate your enthusiasm for our manuscript.

Sincerely,

Dario Mizrachi